# ComENet: Towards Complete and Efficient Message Passing for 3D Molecular Graphs

**Limei Wang**[*]
Texas A&M University
College Station, TX 77843
limei@tamu.edu

**Yi Liu**[*]
Florida State University
Tallahassee, FL 32306
liuy@cs.fsu.edu

**Yuchao Lin**
Texas A&M University
College Station, TX 77843
kruskallin@tamu.edu

**Haoran Liu**
Texas A&M University
College Station, TX 77843
liuhr99@tamu.edu

**Shuiwang Ji**
Texas A&M University
College Station, TX 77843
sji@tamu.edu

## Abstract

Many real-world data can be modeled as 3D graphs, but learning representations that incorporates 3D information completely and efficiently is challenging. Existing methods either use partial 3D information, or suffer from excessive computational cost. To incorporate 3D information completely and efficiently, we propose a novel message passing scheme that operates within 1-hop neighborhood. Our method guarantees full completeness of 3D information on 3D graphs by achieving global and local completeness. Notably, we propose the important rotation angles to fulfill global completeness. Additionally, we show that our method is orders of magnitude faster than prior methods. We provide rigorous proof of completeness and analysis of time complexity for our methods. As molecules are in essence quantum systems, we build the complete and efficient graph neural network (ComENet) by combing quantum inspired basis functions and the proposed message passing scheme. Experimental results demonstrate the capability and efficiency of ComENet, especially on real-world datasets that are large in both numbers and sizes of graphs. Our code is publicly available as part of the DIG library (https://github.com/divelab/DIG).

## 1 Introduction

In machine learning, structured objects such as molecules [14, 53, 52], proteins [9, 16, 35, 39, 30], materials [54], and quantum systems [33, 17] are usually modeled as graphs. Original modeling shows basic connections between units and the resulted data type is known as 2D graphs. Accordingly, 2D graph neural networks (GNNs) have been intensively studied [14, 21, 20, 55, 59, 34] and the message passing scheme [24, 7, 51] is shown effective to realize 2D GNNs. However, in the modern machine learning era, it is increasingly accepted that modeling real-world data like molecules as 2D graphs leads to inborn defects for succeeding learning models. In practice, 3D information is

---

[*]Equal contribution

36th Conference on Neural Information Processing Systems (NeurIPS 2022).

crucial, based on which some important geometries can be derived, such as chemical bond lengths in molecular modeling. This essentially raises the need of a new data type, known as 3D graphs.

Formally, a 3D graph contains the original 2D graph as well as Cartesian coordinates for all nodes. In this work, we follow invariant 3D GNNs [43, 22, 32, 45, 36, 23] that take relative 3D information, like distances and angles, as inputs to networks. Such relative geometries are naturally SE(3)-invariant. The main challenge comes from what geometries should be computed such that 3D information is incorporated completely. A most recent work SphereNet [36] shows that distance, angle, and torsion information is necessary to incorporate more comprehensive 3D information. However, SphereNet is only complete in local neighborhood, failing to achieve global completeness and distinguish a wide range of molecular structures such as conformers. Let $n$ and $k$ denote the number of nodes and the average degree in a 3D graph. Existing methods also exhibit excessive time complexity of $O(nk^2)$ or even $O(nk^3)$, severely preventing their scalability in real-world applications.

In this work, we propose ComENet as a complete and efficient graph neural network for 3D molecular graph learning. We first formally provide the definition of *completeness*. Intuitively, a geometric transformation is considered as complete if it generates distinct representations for any two different 3D graphs. Based on this, we propose a novel message passing scheme by faithfully fulfilling global completeness via the important rotation angles. In addition, we design novel strategies to achieve local completeness, largely reducing the computing complexity to $O(nk)$. To elucidate the merits of the proposed methods, we provide rigorous proof of geometric completeness achieved by our method. Combining the novel message passing scheme and quantum inspired features, ComENet is developed for 3D molecular graphs. We apply ComENet to two large-scale datasets including OC20 and Molecule3D, and a commonly used dataset QM9. Experiments show that ComENet performs similar to existing best methods, but accelerates the training and inference by 6-10 times on various datasets. **We summarize the contributions of ComENet as below**. (i). To our best knowledge, it is the first rigorously complete pipeline for 3D molecular graph learning. Theoretically, it is guaranteed to incorporate 3D information completely without information loss. Practically, it can distinguish all molecular structures in nature. (ii). It is highly efficient. The message passing is shown to be orders of magnitude faster than existing methods in terms of time complexity. (iii). The great capability and efficiency of ComENet allow its scalability to real-world molecule datasets that are large in both numbers and sizes of graphs. (iv). It achieves similar or better performance compared with existing methods, and dramatically accelerates the training and inference by 6-10 times.

**Relations with Prior Work.** SphereNet [36] is a recent method that achieves local completeness with a complexity of $O(nk^2)$. In summary, SphereNet is not complete and not efficient enough for processing large-scale molecular graphs. However, ComENet is complete and much more efficient with a complexity of $O(nk)$. Technically, a primary difference is that, ComENet proposes to use the important rotation angles to achieve global completeness. As a result, ComENet is provably complete as shown in Sec. 2.2. Practically, ComENet is able to identify a wide range of real-world structures such as conformers, achieving completeness at the conformer level, as introduced in Sec. 2.3. In addition, ComENet and SphereNet both use $(d, \theta, \phi)$ in the spherical coordinate system (SCS) to obtain local completeness. Indeed, the fact that $(d, \theta, \phi)$ can specify the location of a point in SCS is widely known. The key difference lies in that, SphereNet operates within 2-hop neighborhood, while ComENet operates within 1-hop neighborhood. When coupled with rotation angles, ComENet can achieve provable completeness with a reduced complexity of $O(nk)$. This different message passing for local completeness in ComENet entails many differences with SphereNet, including building coordinate systems, defining $z$-axis, choosing reference nodes, and computing $(d, \theta, \phi)$, as detailed in Sec. 2.4. All the computing procedures for ComeNet are described in detail in Algorithm 1 of Appendix A.1.

## 2 The Proposed Message Passing Scheme

### 2.1 Notations & Definitions

We first formally define notations and the concept of *completeness* used in this paper.

**Notations.** A 3D graph $G$ can be represented as $G = (V, A, P)$. The node feature matrix $V = [\mathbf{v}_1, \mathbf{v}_2, \cdots, \mathbf{v}_n]^T \in \mathbb{R}^{n \times d_v}$ with each $\mathbf{v}_i \in \mathbb{R}^{d_v}$. The adjacency matrix $A \in \mathbb{R}^{n \times n}$, based on which we additionally define there is an edge $e_{ij}$ if $A[i][j] = 1$. The position matrix $P = [\mathbf{p}_1, \mathbf{p}_2, \cdots, \mathbf{p}_n]^T \in \mathbb{R}^{n \times 3}$, where $\mathbf{p}_i = (x_i, y_i, z_i) \in \mathbb{R}^3$ is the position vector for node $i$ given in the Cartesian coordinate

system (CSC). The relative position $\mathbf{p}_{ij}$ of node $j$ to node $i$ is defined as $\mathbf{p}_{ij} = \mathbf{p}_j - \mathbf{p}_i$. Particularly, throughout this paper, we define $k$ as the average degree for $G$.

We then formally define *completeness* given a geometric transformation $\mathcal{T}$. In particular, we aim at incorporating 3D information in 3D molecular graphs. Hence, our definition of *completeness* is set from the geometric view. Generally, $\mathcal{T}$ maps a 3D graph $G = (V, A, P)$ to a geometric representation with size $m \times h$, where $m$ is the number of transformed geometric features and $h$ is the feature size. $\mathcal{T}$ can be different dependent on different methods, resulting in different $m \in \mathbb{N}^+$ or $h \in \mathbb{N}^+$. For example, SchNet [43] only computes the distance for each edge based on the coordinates of the two nodes connected by this edge. Thus, SchNet maps $G$ to a representation with size $m \times h$, where $m$ is the number of edges in $G$ and $h = 1$. However, such geometric transformation is not complete. We provide the definition of *completeness* as below:

**Definition 1** (Completeness). *For two 3D graphs $G_1 = (V, A, P_1)$ and $G_2 = (V, A, P_2)$, a geometric transformation $\mathcal{T} : (\mathbb{R}^{n \times d_v}, \mathbb{R}^{n \times n}, \mathbb{R}^{n \times 3}) \mapsto \mathbb{R}^{m \times d}$ is considered as complete when*

$$\mathcal{T}(G_1) = \mathcal{T}(G_2) \iff \exists R \in \mathrm{SE}(3), P_1 = R(P_2).$$

Here $\mathrm{SE}(3)$ denotes the Special Euclidean group in 3 dimensions. It include all rotations and translations in 3D [1, 26, 36, 41]. Thus $R$ is a transformation that combines rotation and translation. A rotation transformation can be represented with a $3 \times 3$ rotation matrix, and a translation transformation can be represented by a $3 \times 1$ vector. Matrix form of $\mathrm{SE}(3)$ is provided in Appendix A.2. The reason why we introduce $\mathrm{SE}(3)$ lies in that, a combination of rotation and translation will not change the 3D conformation of a 3D graph. In Def. 1, if $P_1$ and $P_2$ are in the same $\mathrm{SE}(3)$ group, then $G_1$ and $G_2$ would share the same 3D conformation. As a result, $G_1$ and $G_2$ would be the same 3D graph. Intuitively, a complete geometric transformation $\mathcal{T}$ can distinguish any two different 3D graphs. This is to say, as long as two 3D graphs differ in 3D conformations, their outputs from $\mathcal{T}$ would be different.

## 2.2 Global Completeness via Rotation Angles

Existing studies focus on the complete representation learning of local neighborhood. Earlier methods like SchNet [43] and DimeNet [22] cannot achieve local completeness. In a more recent method SphereNet, completeness is guaranteed within edge-based 1-hop local neighborhood, but fails to hold in the whole 3D graph. In this section, we move a step forward to formally fulfill global completeness for a given 3D molecular graph. Particularly, for the purpose of clear illustration, we safely assume local completeness is already obtained by exiting methods like SphereNet.

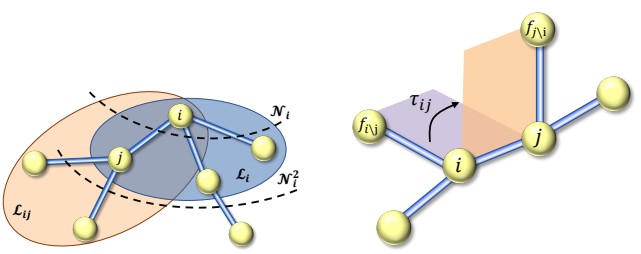

(a) Illustration of a 2-hop local structure.

(b) Illustration of computing the rotation angle for an edge $e_{ij}$.

Figure 1: Illustrations of how to achieve global completeness in our proposed methods.

Without loss of generality, we start by describing our method to attain full completeness within 2-hop neighborhood, as illustrated in Fig. 1(a). Formally, for a center node $i$, we let $\mathcal{N}_i$ and $\mathcal{N}_i^2$ denote two sets of indices of $i$'s 1-hop and 2-hop neighboring nodes, respectively. We also define any node $i$ and its 1-hop neighborhood as a local structure. Then the whole 2-hop neighborhood of node $i$ can be viewed as $1 + |\mathcal{N}_i|$ local structures centered in $i$ and $\mathcal{N}_i$, defined as $\mathcal{L}_i$ and $\mathcal{L}_{ij,j\in\mathcal{N}_i}$, respectively. As shown in Fig. 1(a), $\mathcal{L}_i$ is the local structure centered in $i$, and $\mathcal{L}_{ij}$ is the local structure centered in $j$. Apparently, each local structure $\mathcal{L}_{ij,j\in\mathcal{N}_i}$ shares the common edge $e_{ij}$ with the local structure $\mathcal{L}_i$. Given the complete representation for each local structure, for the structure $\mathcal{L}_i \cup \mathcal{L}_{ij}$, the only remaining degree of freedom is the rotation angle of edge $e_{ij}$, denoted as $\tau_{ij}$. With the rotation angles for all the $|\mathcal{N}_i|$ common edges specified, we can obtain a complete representation for 2-hop neighborhood of node $i$. Achieving completeness beyond 2-hop neighborhood is similar. Overall, after considering rotation angles, the global completeness can be easily guaranteed when it gradually generalizes from $n$- to $(n+1)$-hop neighborhood.

Essentially, each edge in an input graph can be treated as a common edge between different local structures. We reveal how to compute the rotation angle $\tau_{ij}$ for each edge $e_{ij}$ in Fig. 1(b). Specifically,

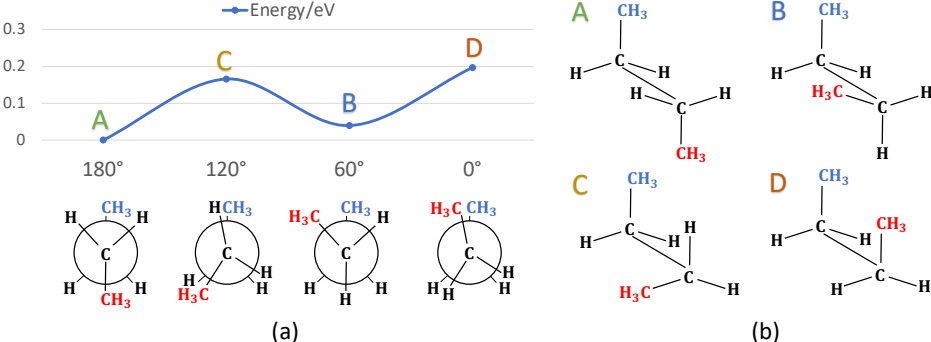

Figure 2: (a). Illustration that the relative conformation energy of butane is a function of the rotation angle of the C-C bond. (b). A 3D view of the four conformers in (a).

we choose two reference nodes whose indices are $f_{i\backslash j}$ and $f_{j\backslash i}$ for node $i$ and $j$, respectively. $f_{i\backslash j}$ denotes the index of $i$'s nearest neighboring node except $j$, and $f_{j\backslash i}$ denotes $j$'s nearest neighboring node except $i$. Then $\tau_{ij}$ for edge $e_{ij}$ is the angle from the plane formed by $f_{i\backslash j}, i, j$ to the plane formed by $i, j, f_{j\backslash i}$. As analyzed previously, the global conformation of the input graph can be identified based on all local structures and rotation angles. As a result, given fixed local structures, the global completeness the input 3D molecular graph is fulfilled by additionally considering the rotation angle of each edge, as introduced in this section. Note that in terms of the selection of reference nodes for computing a rotation angle, we think a selection strategy is valid as long as it is applied to every edge consistently. In this work, for an edge $e_{ij}$, we choose the nearest neighboring nodes for $i$ and $j$ as reference nodes, which are $f_{i\backslash j}$ and $f_{j\backslash i}$, respectively. We apply this selection strategy to all edges in a 3D graph for computing corresponding rotation angles. With such selection strategy, we can prove that our method is complete as shown in Sec. 3.1. We also show in Appendix A.3 that it is easier to prove completeness using nearest neighboring nodes as reference nodes.

## 2.3 Rotation Angles for Conformer Identification

The proposed rotation angles in Sec. 2.2 play a crucial role in identifying some important molecular structures, such as conformers. In nature, a real molecule exits as an ensemble of interconventing 3D structures, known as conformers [5, 4, 19]. Different conformers posses the same 2D molecular graph, but differ in 3D structures. Generally, a conformer for a molecule exits with a certain probability and may exhibit distinct properties [5, 4]. As shown in Fig. 2(a), different conformers for the molecule butane ($C_4H_{10}$) show varying conformation energy. To this end, it is important to design complete 3D GNNs for identifying molecules at the conformer level. From the geometry perspective of view, a conformer distinguishes itself from others mainly through varying rotation angles of chemical bonds [19]. As shown in Fig. 2(b), given the fixed ethyl (-$C_2H_5$) of both sides, the only degree of freedom is the rotation angle of the C-C bond. In literature, the ethyl is formulated as a local 3D graph. Existing studies focus on the complete representation of such local structures, failing to identify the whole 3D graph globally. Essentially, they can only distinguish different molecules, trying to achieve completeness at the molecule level rather that the finer comformer level. By integrating the rotation angles as in Sec. 2.2 into the message passing scheme, our methods can fulfill rigorous completeness at the conformer level and can distinguish all conformers in nature.

## 2.4 Local Completeness with Improved Efficiency

Formally, each node and its local neighborhood can be viewed as a local structure. Existing studies focus on the learning of local structures and SphereNet achieves local completeness. However, SphereNet induces the complexity of $O(nk^2)$, restricting its scalability on large molecules in practice. Here, we design a novel strategy to guarantee local completeness with the computing cost of $O(nk)$.

Specifically, we follow SphereNet and perform on the spherical coordinate system. It is commonly known that the location of each node can be completely determined using the tuple $(d, \theta, \phi)$ in SCS. SphereNet employs the directional message passing (DMP) fashion that operates within 2-hop neighborhood. It first updates messages over edges thus the center edge is $z$-axis in SCS. For node $i$, the computing of the tuple $(d, \theta, \phi)$ involves 2-hop information. However, we view 1-hop neighborhood as a local structure, which requires all strategies in our local completeness to be

different from SphereNet, including building local coordinate systems, defining z-axis, picking reference nodes, and computing $(d, \theta, \phi)$. First, we build a light local coordinate system for any node $i$'s corresponding local structure. Similarly, the center node $i$ serves as the origin. Then $z$-axis is defined as the direction from $i$ to its nearest neighbor $f_i$, and $xz$-plane is further formed by $z$-axis and $i$'s second nearest neighbor $s_i$. Finally, the tuple $(d, \theta, \phi)$ is computed within 1-hop neighborhood with a complexity of $O(nk)$. Particularly, we analyze efficiency versus model expressiveness in Sec. 3.2. We show that compared with the DMP fashion used by SphereNet, our method operating within 1-hop neighborhood hurts the model expressiveness a bit by largely improves the efficiency.

## 2.5 Message Passing Scheme

Based upon global completeness achieved in Sec. 2.2 and improved local completeness introduced in Sec. 2.4, the complete geometric transformation $\mathcal{T}$ required by Def. 1 should be formulated based on a 4-tuple as $(d, \theta, \phi, \tau)$. Specifically, we build such transformation within 1-hop neighborhood, and a 4-tuple is computed for each edge. Hence, for a 3D graph $G = (V, E, P)$, the full expression for $\mathcal{T}$ is $\mathcal{T}(G) = [(d_{ij}, \theta_{ij}, \phi_{ij}, \tau_{ij})]_{i=1,...,n;j \in \mathcal{N}_i} \in \mathbb{R}^{m \times 4}$, where $m$ is the number of edges in $G$. Especially, as $\mathcal{T}$ converts absolute Cartesian coordinates in $P$ to relative information, it is naturally SE(3)-invariant as required in Def. 1. To this end, we formally build our message passing scheme as

$$\mathbf{v}'_i = g \left( \mathbf{v}_i, \sum_{j \in \mathcal{N}_i} f \left( \mathbf{v}_j, d_{ij}, \theta_{ij}, \phi_{ij}, \tau_{ij} \right) \right), \tag{1}$$

where $g$ and $f$ can be implemented by neural networks or mathematical operations. Intuitively, our message passing is established in 1-hop local neighborhood and all edges connecting regions beyond. Essentially, $d_{ij}$, $\theta_{ij}$, and $\phi_{ij}$ specify the 1-hop local neighborhood, and $\tau_{ij}$ determines the orientation of the local neighborhood. By doing this, the complete representation for a whole 3D molecular graph is eventually achieved. The formulas of computing of $d_{ij}$, $\theta_{ij}$, and $\phi_{ij}$, and $\tau_{ij}$ are shown in Algorithm 1 in Appendix A.1, along with detailed description of the complexity of $O(nk)$. Overall, our formal analysis in Sec. 2.2, Sec. 2.3, and Sec. 2.4 lead to the proposed message passing scheme defined in Eq. 1. It is the first fully complete scheme with great efficiency of $O(nk)$. We also provide rigorous proof on completeness and analysis on efficiency of our message passing in Sec. 3. Note that to achieve efficiency, our message passing scheme adopts a novel strategy that computes all the needed geometries within 1-hop neighborhood. Hence, it can not be directly applied to existing architectures built in 2-hop neighborhood, such as DimeNet++ [32] and SphereNet [36]. To this end, we design a new network to implement the proposed message passing scheme, as detailed in Sec. 4.

## 3 Merits of Our Methods

### 3.1 Geometric Completeness

**Proposition 1.** *For a strongly connected 3D graph $G = (V, E, P)$, its geometric transformation $\mathcal{T}(G) = [(d_{ij}, \theta_{ij}, \phi_{ij}, \tau_{ij})]_{i=1,...,n;j \in \mathcal{N}_i}$ is complete.*

*Proof.* We employ mathematical induction and assume the number of nodes in a 3D graph is $n$. Note that the 3D graph we consider is strongly connected, which means that there exist a path between any two nodes in the graph. All the molecules in nature can be constructed as strongly connected graphs.

Base case: It is obvious that the 3D structure of $G$ can be identified when $n = 1, 2$. Hence, we let $n = 3$ be the base case, where the completeness can be achieved by only considering $d$ and $\theta$ in $\mathcal{T}$.

Inductive hypothesis: The claim that $\mathcal{T}$ is complete holds for the node numbers of $n$ up to $k \geq 3$.

Inductive step: Let $n = k + 1$. Without loss of generality, among the existing $k$ nodes, we safely assume $i$ and its neighboring nodes $c_{k=1,2,...}$ form the local region of interest. Then $j$ is the index of the newly $(k + 1)$-th node connected to the center node $i$. To show global completeness of the whole graph, based on Def. 1, we only need to prove that the relative position of the new node $j$ is uniquely determined given $\mathcal{T}$. We propose the following lemma for this as

**Lemma 1.** *Assume a strongly connected 3D graph $G = (V, E, P)$ with more than 2 nodes is fully identified. If a new node $j$ is connected to a node $i$ of $G$ following the geometric transformation $\mathcal{T}(G) = [(d_{ij}, \theta_{ij}, \phi_{ij}, \tau_{ij})]_{i=1,...,n;j \in \mathcal{N}_i}$, then $\boldsymbol{p}_{ij}$ is uniquely determined.*

The proof of Lemma 1 is provided in Appendix A.3. With Lemma 1 successfully proved, we show that such geometric transformation $\mathcal{T}$ can determine a unique 3D graph. Hence, the *if* condition in Def. 1 holds. In addition, as $\mathcal{T}$ renders purely relative 3D information like distance and angle, it's naturally SE(3) invariant. Hence, the *only if* condition in Def. 1 holds. Overall, based on Def. 1, we complete the proof of Prop. 1. $\qquad\square$

Intuitively, since a molecular graph is strongly connected, two arbitrary nodes are connected by at least one path. Hence, starting from the existing structure, we can restore the relative position of any new node step by step along a path with finite length. As a result, geometric completeness for the whole 3D graph with any number of nodes can be guaranteed in our message passing scheme. Theoretically, 3D information of the input 3D molecular graph is fully captured without information loss. In practice, our method can distinguish all structures in nature.

## 3.2 Efficiency

**Efficiency versus Model Expressiveness:** Our method induces the complexity of $O(nk)$ by operating within 1-hop neighborhood. Existing methods, such as like DimeNet and SphereNet, employ the DMP fashion that update edges within 2-hop neighborhood, inducing the complexity of $O(nk^2)$. Notably, DMP [48, 57, 22, 36] incorporates 2-hop information in one single layer but $(n+1)$-hop when stacking $n$ layers. However, stacking $n$ proposed message passing layers is already able to incorporate information from $n$-hop neighborhood. Obviously, compared with a network containing several DMP layers, a network with the same number of our proposed message passing layers merely hurts the model expressiveness a bit, but significantly improves the model efficiency.

**Efficiency via Less Torsion Angles:** In addition, our method achieves completeness by computing $O(nk)$ torsion angles, which is efficient especially compared with methods including Gasteiger et al. [23], Adams et al. [1], Ganea et al. [19]. Given a scenario where a global region is the union of two local regions with $n_1$ and $n_2$ nodes. As introduced in Sec. 2.2 and Sec. 2.4, our method computes the same number of torsion angles as nodes in a local region, then employs a rotation angle (torsion angle essentially). Hence, the number of torsion angles that our method computes is $n_1 + n_2 + 1$. Ganea et al. [19] computes a torsion angle based on one pair of nodes, each of which is from a separate local region. Hence, the number of torsion angles needed is $n_1 \times n_2$. Apparently, our method reduces the number of torsion angles from $O(nk^3)$ to $O(nk)$, which is significant considering the computing of torsion is excessively expensive.

## 4 ComENet

Based upon the message passing scheme introduced in Sec. 2.5, we propose the complete and efficient graph neural network (ComENet) as shown in Fig. 3. Existing invariant 3D GNN methods [42, 22, 32, 36, 23, 44] share the similar architecture pipeline, which contains an input block, several interaction blocks, and an output block. Our ComENet also follows such architecture fashion along with several novel components, such as self-atom layers and specifically designed local and global graph convolution layers, to better fulfill our proposed message passing scheme in Eq. 1. Generally, ComENet consists of an embedding layer, multiple interaction layers, a self-atom layer, and a pooling layer. To be in line with the message passing scheme in Sec. 2.5, we take the updating process for node

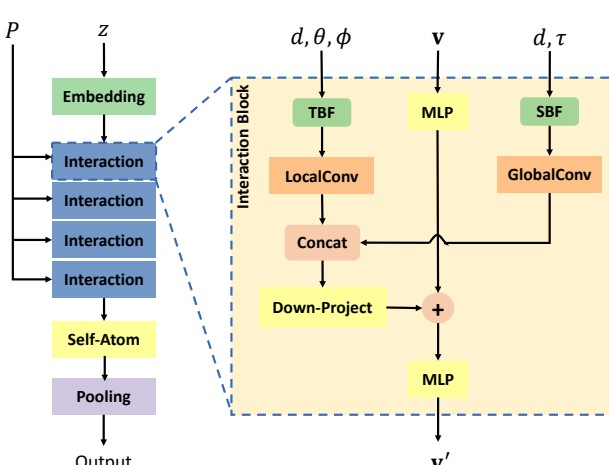

Figure 3: Illustration of ComENet with an overview (left) and the interaction layer (right). TBF and SBF denote the basis functions for tuples $(d, \theta, \phi)$ and $(d, \tau)$. LocalConv and GlobalConv denote the proposed local and global convolution layers. Concat is the concatenation operation and Down-Project is a linear layer to reduce feature dimensions. + denotes the element-wise sum operation.

$i$'s feature vector $\mathbf{v}_i$ as an example to describe the network. In practice, feature vectors for all the nodes in a graph are updated simultaneously. In particular, we omit all indices in Fig. 3 and below for clear presentation. Specifically, *embedding Layer* converts atom type z to an initial node feature vector $\mathbf{v}$ via learnable atom type embeddings [43, 22]. *Interaction Layer* updates node feature vector $\mathbf{v}$ based on features of the neighboring nodes and geometric features $(d, \theta, \phi, \tau)$ in Eq. 1 using the local and global graph convolution layers. A detailed description of interaction layer is provided in Appendix A.4. *Self-atom layer* is used to update each node feature and project the feature dimension into 1. And *pooling layer* is a sum-pooling performing on all node features to obtain final predictions.

## 5 Related Work

We consider how to represent 3D information in 3D molecular graphs [3, 10]. One category of methods are *equivariant 3D GNNs* that directly use coordinates in the CSC as inputs to networks [49, 2, 18, 44, 8]. These methods are efficient but suffer from several setbacks. Firstly, each network component needs to be carefully designed to be rotation equivariant of input graphs. Secondly, the reason why this category of methods are efficient lies in that they only use the type-1 basis in Spherical harmonics, which is an approximation essentially. It is proved in [49] that theoretically, $l$ is infinite in terms of type-$l$ basis. In practice, $l$ should be at least 2 for achieving satisfactory performance. Type-1 basis essentially coerces the conv kernel to be in a narrow learning space, which imposes a hard constraint on the network capability. However, when using type-2 basis, the conv kernel can be more expressive while the efficiency issue would emerge as a new bottleneck. Thirdly, the performance of such equivariant GNNs is shown to be worse than invariant 3D GNNs [36].

In this work, we follow *invariant 3D GNNs* to inherit the merit of SE(3)-invariance by investigating relative 3D information, which is used in both representation learning tasks [43, 22, 32, 45, 36, 23] and coordinates generation tasks [19, 46, 47, 56, 30, 6]. Since the relative information is SE(3)-invariant of input 3D graphs, the employed networks favorably achieve the invariance merit. We focus on the 3D graph learning problem and existing methods either capture partial 3D information or suffer from high computational cost. For example, SchNet [43] only considers distance information and DimeNet [22] further incorporates angles between bonds. They both integrate incomplete 3D information that the network capacity is limited in practice. SphereNet [36] generates approximate complete 3D representations by using distance, angle, and torsion information but the complexity is $O(nk^2)$. GemNet [23] is based on quadruplets of nodes, which is more expensive. Our objective is to build a fully complete 3D graph net with a much lower computational budget.

There also exist some methods in literature using *both absolute and relative 3D information* [25, 28, 58]. In this work, we only use relative information as input to avoid inherent limitations of equivariant 3D GNNs. Moreover, we design a novel message passing such that the computational cost is comparable with that of equivariant 3D GNNs.

## 6 Experiments

We examine the power and efficiency of ComENet on two large-sacle datasets including Open Catalyst 2020 (OC20) [12] and Molecule3D [56], and the mostly commonly used datatset QM9 [40]. The statistics of three datasets are provided in Table 1. Detailed descriptions of the

Table 1: Statistics of the datasets.

| Dataset | OC20 | Molecule3D | QM9 |
|---|---|---|---|
| # Graphs | 660,010 | 3,899,647 | 130,831 |
| Split Type | Pre-defined | Random/Scaffold | Random |
| Split Ratio | 70:15:15 | 6:2:2 | 84:8:8 |
| # Nodes/Graph | 77.75 | 29.11 | 18.02 |

datasets are provided in Appendix A.5. In particular, the Molecule3D contains about 4 million 3D molecular graphs, and the OC20 has the average graph size of 77.75. Baseline methods include GIN-Virtual [27], CGCNN [54], SchNet [43], PhysNet [50], MGCN [37], DimeNet [22], DimeNet++ [32], SphereNet [36], PaiNN [44], GemNet [23]. Unless otherwise specified, the values for baseline methods are taken from the referred papers. For the ComENet, we use data loader in the PyTorch Geometric library [15] to load the datasets. All the models are trained using the Adam optimizer [31] and the optimal hyperparameters are obtained on validation sets using grid search. Experimental setup and search space for all models are provided in Appendix A.6. Code is integrated in the DIG library [34] and available at `https://github.com/divelab/DIG`.

Table 2: Results on IS2RE including computing cost in training&inference and performance in terms of energy MAE and the percentage of EwT of the ground truth energy. Training time is the average time per epoch during training using 1 GPU. Performance is reported for models trained on the All training dataset. The best performance is shown in bold and the second best is shown with underlines.

| | Time | | Energy MAE [eV] ↓ | | | | | EwT ↑ | | | | |
|---|---|---|---|---|---|---|---|---|---|---|---|---|
| Model | Train | Infer. | ID | OOD Ads | OOD Cat | OOD Both | Average | ID | OOD Ads | OOD Cat | OOD Both | Average |
| CGCNN | 18min | 1min | 0.6203 | 0.7426 | 0.6001 | 0.6708 | 0.6585 | 3.36% | 2.11% | 3.53% | 2.29% | 2.82% |
| SchNet | 10min | 1min | 0.6465 | 0.7074 | 0.6475 | 0.6626 | 0.6660 | 2.96% | 2.22% | 3.03% | 2.38% | 2.65% |
| DimeNet++ | 230min | 4min | 0.5636 | 0.7127 | 0.5612 | 0.6492 | 0.6217 | 4.25% | 2.48% | 4.40% | 2.56% | 3.42% |
| GemNet-T | 200min | 4min | 0.5561 | 0.7342 | 0.5659 | 0.6964 | 0.6382 | 4.51% | 2.24% | 4.37% | 2.38% | 3.38% |
| SphereNet | 290min | 5min | 0.5632 | 0.6682 | 0.5590 | 0.6190 | 0.6023 | **4.56%** | 2.70% | **4.59%** | 2.70% | **3.64%** |
| ComENet | 20min | 1min | **0.5558** | **0.6602** | **0.5491** | **0.5901** | **0.5888** | 4.17% | **2.71%** | 4.53% | **2.83%** | 3.56% |

## 6.1 OC20

The Open Catalyst 2020 (OC20) dataset is a newly released large-scale dataset with millions of DFT relaxations to model and discover catalysts. In this work, we focus on Initial Structure to Relaxed Energy (IS2RE) task, which is the most common task in catalyst discovery. Descriptions of the data and tasks are provided in Appendix A.5. The ground truth of the test set is not publicly available, therefore, we compare the results of different methods on the validation set. The evaluation metrics include the energy MAE and the percentage of Energies within a Threshold (EwT) of the ground truth energy. The values for the baseline methods are taken from Chanussot et al. [11], Liu et al. [36]. Notably, we aim to predict relaxed energy directly from initial structure and do not compare with some methods using relaxation [12], trajectory information, or relaxed structures. Using relaxation [12, 23, 45] is computationally expensive during prediction while the relaxation trajectory and relaxed structures [25, 58] are hard to obtain in practice.

Table 2 shows that ComENet outperforms all the baseline methods in terms of energy MAE, which is also used as the main evaluation metric in the Open Catalyst Challenge [13]. ComENet achieves best performance on two splits and the second best on the other two splits in terms of EwT. Specifically, ComENet reduces the average energy MAE by 0.0135, which is 2.2% of the second best model. Note that ComENet achieves the best results on the OOD Both split in terms of both energy MAE and EwT. In practice, it is common that test data is in the different domain with the training data. Hence, OOD Both can test the generalization capability of learning models. More importantly, ComENet is much more efficient than methods like DimeNet++ and SphereNet. For example, SphereNet needs 5 hours per epoch while ComENet only requires 20 minutes using the same computing infrastructure (NVIDIA RTX A6000 48GB). Overall, compared with existing best methods, ComENet achieves better performance and largely reduces training by at least 10 times.

## 6.2 Molecule3D

The Molecule3D dataset [56] is a newly proposed large-scale dataset, including around 4 million molecules with precise ground-state 3D information derived from DFT and molecular properties. We focus on the prediction of the HOMO-LUMO gap as it is one of the most practically-relevant quantum chemical properties of molecules. A detailed description of the Molecule3D dataset is provided in Appendix A.5. As this is a newly proposed dataset, we run baseline methods including GIN-Virtual [27],

Table 3: Comparisons between ComENet and other models in terms of computing cost and HOMO-LUMO gap MAE on Molecule3D for both random and scaffold splits. Train time is the average training time per epoch.

| | Time | | MAE | |
|---|---|---|---|---|
| Model | Train | Inference | Random | Scaffold |
| GIN-Virtual | 15min | 2min | 0.1036 | 0.2371 |
| SchNet | 14min | 3min | 0.0428 | 0.1511 |
| DimeNet++ | 133min | 16min | 0.0306 | 0.1214 |
| SphereNet | 182min | 28min | 0.0301 | 0.1182 |
| ComENet | 22min | 3min | 0.0326 | 0.1273 |

SchNet [43], DimeNet++[32] and SphereNet [36], among which GIN-Virtual is a powerful baseline for 2D graphs while the others are for 3D graphs. All the models are trained using the same computing infrastructure (Nvidia GeForce RTX 2080 Ti 11GB).

Table 4: Comparisons between ComENet and other models in terms of MAE and the overall mean std. MAE on QM9.

| Property | Unit | SchNet | PhysNet | MGCN | DimeNet | DimeNet++ | PaiNN | SphereNet | ComENet |
|---|---|---|---|---|---|---|---|---|---|
| $\mu$ | D | 0.033 | 0.0529 | 0.0560 | 0.0286 | 0.0297 | 0.012 | 0.0245 | 0.0245 |
| $\alpha$ | $a_0{}^3$ | 0.235 | 0.0615 | 0.0300 | 0.0469 | 0.0435 | 0.045 | 0.0449 | 0.0452 |
| $\epsilon_{HOMO}$ | meV | 41 | 32.9 | 42.1 | 27.8 | 24.6 | 27.6 | 22.8 | 23.1 |
| $\epsilon_{LUMO}$ | meV | 34 | 24.7 | 57.4 | 19.7 | 19.5 | 20.4 | 18.9 | 19.8 |
| $\Delta\epsilon$ | meV | 63 | 42.5 | 64.2 | 34.8 | 32.6 | 45.7 | 31.1 | 32.4 |
| $\langle R^2 \rangle$ | $a_0{}^2$ | 0.073 | 0.765 | 0.110 | 0.331 | 0.331 | 0.066 | 0.268 | 0.259 |
| ZPVE | meV | 1.7 | 1.39 | 1.12 | 1.29 | 1.21 | 1.28 | 1.12 | 1.20 |
| $U_0$ | meV | 14 | 8.15 | 12.9 | 8.02 | 6.32 | 5.85 | 6.26 | 6.59 |
| $U$ | meV | 19 | 8.34 | 14.4 | 7.89 | 6.28 | 5.83 | 6.36 | 6.82 |
| $H$ | meV | 14 | 8.42 | 14.6 | 8.11 | 6.53 | 5.98 | 6.33 | 6.86 |
| $G$ | meV | 14 | 9.4 | 16.2 | 8.98 | 7.56 | 7.35 | 7.78 | 7.98 |
| $c_v$ | $\frac{cal}{mol\ K}$ | 0.033 | 0.028 | 0.038 | 0.025 | 0.023 | 0.024 | 0.022 | 0.024 |
| std. MAE | % | 1.76 | 1.37 | 1.86 | 1.05 | 0.98 | 1.01 | 0.91 | 0.93 |

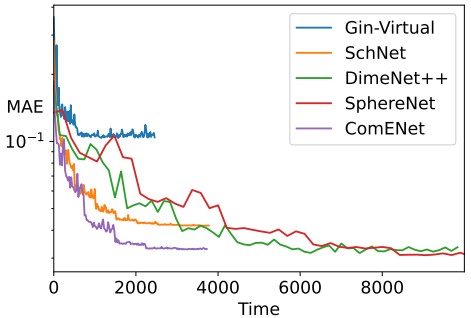

Figure 4: Total training time for different methods on Molecule3D.

Table 3 shows that ComENet dramatically reduces training time by 6-9 times compared with DimeNet++ and SphereNet, and costs similar time as GIN-virtual and SchNet that only considers distance information. In terms of performance, our ComENet performs much better than SchNet for both random and scaffold splits with similar time and computing costs, but a little worse than DimeNet++ and SphereNet. This may be due to the molecules are relatively small in Molecule3D compared with OC20, the structures that our complete strategy can distinguish may not exist in the dataset. However, considering the comparable performance and high efficiency, our ComENet is more practically useful than other methods on such large datasets. In addition, Fig. 4 also shows ComENet either converges much faster in terms of total training time or performs much better compared with other baselines.

## 6.3 QM9

The QM9 dataset is a widely used dataset for predicting various properties of molecules. The evaluation metrics include the MAE for each property and the overall mean standardized MAE (std. MAE) for all the 12 properties. A detailed description of the dataset is provided in Appendix A.5. Notably, we do not list results for PPGN [38] and Cormorant [2] since they use different train/val/test sizes. Table 4 shows that ComENet is much better than the methods operating in 1-hop neighborhood like SchNet, PhyNet, and MGCN. Compared with DimeNet, DimeNet++, PaiNN, and SphereNet, ComENet achieves similar results on all properties and the overall std. MAE. Consistently, compared with methods operating in 2-hop neighborhood like DimeNet, DimeNet++, and SphereNet, ComENet is much more efficient.

## 6.4 Ablation Study for Identifying Conformers

As mentioned in Sec. 2.3, rotation angles are the main difference between different conformers [19, 29]. We investigate the contribution of our proposed rotation angles $\tau$ to demonstrate the effectiveness of our global complete representations. We conduct experiments on the OC20 dataset since there exist different conformers for molecules in this dataset.

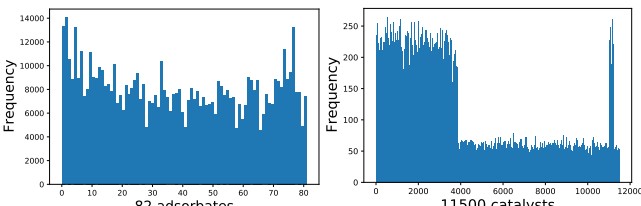

Figure 5: Distributions of adsorbates and catalysts in OC20. For y-axis, frequency counts the number of conformers for each individual adsorbate and catalyst.

Specifically, there are 660,010 data samples in the OC20 dataset (IS2RE), where each sample is a combination of two parts, namely, adsorbate and catalyst. There are 82 adsorbates and 11,500 catalysts used in the datasets. Each adsorbate inevitably corresponds to different conformers in the dataset, and it is similar to catalyst. We show the number of conformers for each adsorbate and each catalyst in Fig. 5. We remove the rotation angle $\tau$ from ComENet and denote it as "ComENet w/o $\tau$". The results in Table 5 show that removing rotation angles $\tau$ can harm the performance of ComENet, demonstrating the effectiveness of our global complete representations for identifying conformers.

Table 5: Comparisons between ComENet and the model without rotation angles $\tau$ on OC20.

| Model | Energy MAE [eV] $\downarrow$ | | | | | EwT $\uparrow$ | | | | |
|---|---|---|---|---|---|---|---|---|---|---|
| | ID | OOD Ads | OOD Cat | OOD Both | Average | ID | OOD Ads | OOD Cat | OOD Both | Average |
| ComENet | **0.5558** | **0.6602** | **0.5491** | **0.5901** | **0.5888** | **4.17%** | **2.71%** | **4.53%** | **2.83%** | **3.56%** |
| ComENet w/o $\tau$ | 0.5585 | 0.6851 | 0.5574 | 0.6186 | 0.6049 | 4.13% | 2.65% | 4.13% | 2.75% | 3.42% |

# 7 Conclusions, Limitations, Outlook, and Societal Impacts

3D information is crucial for 3D molecular graph learning. Existing methods either learn partial 3D information or induce high time complexity. We propose ComENet that is both complete in incorporating 3D information and efficient with time complexity of $O(nk)$. Particularly, we propose the novel rotation angles to fulfill global completeness. ComENet can generalize to large-scale datasets, accelerating training and inference by 6-10 times with superior or comparable performance. Even though ComENet is the first complete and the most efficient 3D GNN model, there exists one major limitation, which is not only for ComENet but also for existing 3D GNNs. Basically, existing 3D GNN models are centered on the fact that 3D information is given in data. However, acquiring 3D information itself is difficult and expensive in practice. Current methods rely on experiments or DFT-based computing, which is extremely time-consuming. It would be significant that machine learning models can be developed to tackle this problem. Looking forward, we can derive two directions for fulfilling such objective. Firstly, we can study to generate 3D graphs either from 2D graphs or from scratch by developing generative models, such as VAE, flow and diffusion models. Especially, in some real-world applications like drug discovery, 2D molecules are usually not given. This raises the need of developing new generation methods from scratch. Secondly, we can target at a research case where we have a minimal set of training data with 3D information, but the vast unseen data or new data lack such 3D information. We may develop novel contrastive learning components to force correspondence and consistency between 2D graphs and their 3D geometric views, then integrate such components into end-to-end learning systems for application based on 2D graph data. ComENet can facilitate a plethora of important real-world applications, such as drug discovery and material discovery. It can be used in several research domains including quantum chemistry and physics, material sciences, molecular dynamics simulations, etc. Any negative societal impact associated with those applications and domains can be applied to our method.

## Acknowledgments and Disclosure of Funding

This work was supported in part by National Science Foundation grant IIS-1908220 and National Institutes of Health grant U01AG070112.

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
