# OpenReview forum: "ComENet: Towards Complete and Efficient Message Passing for 3D Molecular Graphs"
_NeurIPS.cc/2022/Conference — NeurIPS 2022 Accept_

### Official Review · Reviewer_SSbD · 2022-07-11

**Rating:** 6
**Confidence:** 3
**Soundness:** 3 good
**Presentation:** 3 good
**Contribution:** 3 good

**Summary:**

The paper proposes a new neural network architecture to learn representations of 3D graphs, primarily
oriented towards applications to molecules.  The method uses a message passing scheme
operating in 1-hop neighborhoods of each node to propagate information throughout the graph efficiently.
It is said to guarantee global completeness, i.e., distinct conformers of the same molecule yield
distinct representations.

**Questions:**

p.2, line 83: need a reference for SchNet at first mention; how is the distance generated for an edge?

p.3, line 89: missing definition/reference for SE(3).

p.3, line 91: what is included in "differences in structures"? what does "structure" refer to?

p.3, line 94: need a reference for DimeNet at first mention.

p.3, line 95: how large is a local neighborhood with guaranteed completeness by SphereNet?

p.6, Figure 3: how is the single-atom unit shown in the figure applied to the entire molecule?
It will be useful to show an example with, say, one of the conformers in Figure 2(b).
Is each atom in a molecule to be passed through it one by one in turn?  Or are they concatenated
to serve as input to another network for a "downstream" prediction task?

p.8, p.9: the list of Properties in Tables 4 and 5 requires spelled out names and also references for their definitions.



**Limitations:**

The datasets used in the experiments contain relatively small molecules (# nodes/graph < 100).
Is it possible to directly apply this method to much larger molecules?
Some comments on the challenges and limitations there are needed.

**Strengths And Weaknesses:**

The proposal is sound and appears to have good reference value for practitioners.
The method is tested with large public datasets and compared to prior art to demonstrate its effectiveness.

At places the writing of the paper requires some clarification.

---

> ### Author Response · Authors · 2022-08-02
> **Clarified some parts; applied directly to much larger molecules (proteins), achieved good performance using only one GPU (2nd part)**
>
> > Application to much larger molecules.
>
> Thanks for your suggestion to test our method on much larger molecules.
>
> - **Proteins** are highly complex macromolecules with **hundreds or even thousands of amino acids**. We treat each amino acid as a node in a protein graph and directly apply our method to a protein classification task used in [2].
> This task classifies proteins based on the enzyme-catalyzed reaction according to the Enzyme Commission (EC) number. There are a total of 37,428 proteins in this dataset. **The average number of amino acids in each protein is 300, and the largest number of amino acids is 3615. Therefore, the graph size in this data is much larger.**
> - We simply treat each amino acid as a node and apply our method to this protein dataset. The accuracy and the number of parameters of our method and some baseline methods are listed in the table below. Note that the results for baseline methods are copied from [2].
>      |Method|Accuracy|\# Parameters|
>     |----|----|----|
>     |ComENet|86.5%|1.4M|
>     |SphereNet|Out Of Memory|1.4M|
>     |Method in [4]|74.3%|31.7M|
>     |Method in [3]|78.8%|6.0M|
>     |IEConv [2]|87.2%|9.8M|
>
> - From this table, we can see that even though we just simply apply our method to this protein data, our result is still very good. ComENet can outperform most existing protein learning baseline methods. Compared to [2], our method has a small number of parameters. In addition, **our ComENet is trained using only one  NVIDIA GeForce RTX 2080 Ti 11GB GPU.** It takes about 2 mins per epoch and 10 hours to converge. But SphereNet cannot be applied to this data since the training requires a lot of memory. IEConv is specifically designed for protein learning considering heavy domain knowledge, and it needs much more parameters. Our ComENet can still achieve comparable performance with much higher efficiency. This demonstrates the great power and efficiency of our ComENet in learning 3D structures for real-world large data.
> - **We can put the results on proteins in the Appendix of the paper if you suggest so.**
>
> **In summary, this experiment result shows that our ComENet can be directly applied to much larger molecules like proteins with good performance and efficiency.**
>
> We sincerely thank you for your time! We look forward to your reply and further discussions, thanks!
>
> [1] Gilmer, Justin, et al. "Neural message passing for quantum chemistry." ICML 2017.
> [2] Hermosilla, Pedro, et al. "Intrinsic-extrinsic convolution and pooling for learning on 3d protein structures." ICLR, 2021.
> [3] Derevyanko, Georgy, et al. "Deep convolutional networks for quality assessment of protein folds." Bioinformatics 34.23 (2018): 4046-4053.
> [4] Bepler, Tristan, and Bonnie Berger. "Learning protein sequence embeddings using information from structure." ICLR, 2019.

---

> ### Author Response · Authors · 2022-08-02
> **Clarified some parts; applied directly to much larger molecules (proteins), achieved good performance using only one GPU (1st part)**
>
> Thanks a lot for your constructive comments! We have revised the manuscript accordingly and also provide responses here.
>
> > Clarify some parts.
> - We have revised paper and added the reference for **SchNet** at first mention. SchNet is one of the earliest studies to learn representations and predict properties of 3D graphs. Each node in 3D graphs has its 3D coordinates. Therefore, for each edge, SchNet computes the distance between the two nodes.
>
> - **SE(3)** is Special Euclidean group in 3 dimensions. It includes arbitrary combinations of rotations and translations in 3D. Thus $R \in SE(3)$ is a transformation that combines rotation and translation. A rotation transformation can be represented with a $3 \times 3$ rotation matrix, and a translation transformation can be represented by a $3 \times 1$ vector. **SE(3) is important in molecular learning**, and the reason why we introduce it here is straightforward: **When we rotate a 3D molecule, the coordinates of all nodes would change, BUT it’s still the same molecule!** Translation is the same. We need to introduce **SE(3)** to try to avoid violating this nature. **We added several statements after Definition 1 to make it clear to readers.**
>
> - `two 3D graphs differ in structures`:  Here **structure** means 3D conformation (3D shape) of a 3D graph. More specifically, for any two 3D graphs $G_1=(V, A, P_1)$ and $G_2=(V, A, P_2)$, they have the same structure if and only if $\exists R\in SE(3), P_1=R(P_2)$. Otherwise, these two 3D graphs differ in structures. If using some symbols as an example, we say that $\uparrow$ and $\downarrow$ have the same **structure** since they can be matched only with rotation and translation transformations. **However, to avoid confusion, we changed `structure` to `3D conformation`.**
>
> - We have revised paper and added the reference for **DimeNet** at first mention.
>
> - **How large is a local neighborhood with guaranteed completeness by SphereNet?** can only guarantee completeness within edge-based 1-hop local neighborhood. Edge-based 1-hop local neighborhood involves 2-hop nodes. Hence, we can also say it's node-based 2-hop local neighborhood. SphereNet only considers local spherical coordinates, but ignores the rotation angles we proposed in this paper. **We added details to the corresponding lines to make this clearer.**
>
> - **Figure 3, how to apply to an entire molecule?** Thanks for your question. In practice, **all atoms in a molecule are passed to the model simultaneously.** Starting from the first message passing paper [1], all studies based on the message passing framework illustrate the ideas through showing the update process for each node $i$. **This is basically for easy and simple illustration.** In implementation, **all the atoms features are updated at the same time**, and learnable parameters are shared by all atoms. Overall, since each atom has a feature vector, then feature vectors for all atoms form a feature matrix. Operations on this matrix are equivalent to operations on each feature vector parallelly. **We added a statement at the end of the first paragraph in Sec 4 of the paper to clarify this.**
> After passing to the model, each atom has its updated feature vector. We then use a pooling layer (in Figure 3) to sum feature vectors of all atoms to one feature vector. Thus for each molecule, we achieve one feature vector. This feature vector is finally used for downstream prediction tasks (for final prediction).
>
> - **Property names and definitions.** For the QM9 dataset, it includes geometric, energetic, electronic, and thermodynamic properties for 134k stable small organic molecules. The twelve properties are dipole moment ($\mu$), isotropic polarizability ($\alpha$), highest occupied molecular orbital energy ($\epsilon_\text{HOMO}$), lowest unoccupied molecular orbital energy ($\epsilon_\text{LUMO}$), gap between $\epsilon_\text{HOMO}$ and $\epsilon_\text{LUMO}$, electronic spatial extent ($\left< R^2 \right>$), zero point vibrational energy (ZPVE), internal energy at 0K ($U_0$), internal energy at 298.15K ($U$) , enthalpy at 298.15K ($H$), free energy at 298.15K ($G$) , and heat capavity at 298.15K ($c_\text{v}$). **We revised the paper and added a paragraph to describe details of the dataset in Appendix A.6.**
>
> We continue at the 2nd part as below:

---

> ### Author Response · Authors · 2022-08-08
> **Author’s follow-up to reviewer SSbD, one day before author-reviewer conversation ends**
>
> Dear Reviewer SSbD,
>
> Thanks again for your valuable comments and suggestions in your initial review, which helps improve our work a lot. Regarding your main concerns on `application to larger molecules`,  `clarification in writing`, and `lack of limitations`, we have conducted substantial experiments and also revised the paper heavily in our rebuttal on August 1st. Could you please check at your earliest convenience? Thanks!
>
> About `application to much larger molecules`, we do believe this comment is extremely valuable and critical in demonstrating the generalization capability of our model. Thanks again for your suggestion! As shown in our rebuttal and revised paper, we directly applied our method to a protein dataset by treating each amino acid as a node. **The average number of amino acids in each protein is 300, and the largest number of amino acids is 3615**. Therefore, the graph size in this dataset is much larger. Our result is very good compared to some baseline methods. In addition, our method has only a small number of parameters and can be run using only one GPU.
>
> About `some clarifications`, we have revised our paper heavily to make it clearer to readers.
>
> About  `lack of limitations`, we added a long paragraph to state the limitations, and we also provided an outlook for future directions.
>
> We hope that you could reply to our rebuttal and consider raising your score if we do have addressed your concerns. Also, please let us know if there are any additional concerns or feedback. Thank you!
>
> Sincerely,
> Authors

---

> ### Author Response · Authors · 2022-08-09
> **Dear Reviewer SSbD**
>
> Dear Reviewer SSbD,
>
> Thanks again for your valuable comments and suggestions! we posted our rebuttal on August 1st and sent the summary on August 8th. we have conducted substantial experiments and also revised the paper heavily to address your concerns. Could you reply to our rebuttal and consider raising your score if we do have addressed your concerns? Also, please let us know if there are any additional concerns or feedback. Thank you!
>
> Sincerely,
> Authors

---

### Official Review · Reviewer_eaXa · 2022-07-11

**Rating:** 6
**Confidence:** 3
**Soundness:** 3 good
**Presentation:** 2 fair
**Contribution:** 3 good

**Summary:**

This paper proposes ComENet designed for the representation learning on 3D molecular graphs. The framework involves the message passing schemes to capture local completeness and the rotation angle to help capture the global completeness. The paper shows the computational advantage of the proposed model over the existing one and the experimental results show its effectiveness.

**Questions:**

I hope the author could address the concerns raised in the above section.

**Limitations:**

The paper does not discuss the limitation and the social negative impact.

**Strengths And Weaknesses:**

Strengths:
1. Instead of the intuition, the paper defines the global completeness and provides theoretical ground for the model to capture the global completeness.
2. The experiments show the effectiveness of the model.

Weakness:
1. There are a lot of places that need to be further clarified:
    a) Line 33, in O(nk^2), what are n and k? They should be explicitly defined before being used.
    b) Line 82: what is m in m x d?
    c) Line 88: What are the function R() and SE(3)? I would suggest the author to add those in the main text of the appendix to make the paper more readable.
    d) Proposition 1: what is the definition of "strongly connected"? Does it mean the line 199 that two arbitrary nodes are connected by at least one path?
    e) Line 245-246: it would be more clear to say "node feature vector v".
    f) Line 113: What is the definition of L_{ij, j\in N_i}? It seems that its notation is different from previously defined L_i.
2. A tiny typo: line 94: It should be achieve rather that achieves.
3. I'm confused about how to rotation angle. In the example of Figure 1 (b), the angle is calculated as the angle between two planes indicated in the figure. But why is it computed between plane f_{i\j}-i-j and f_{j\i}-i-j rather than other planes that contain the edge i-j (e.g., the plane formed by i, j and the node at the top right corner)? How to choose plane to calculate the angle?

---

> ### Author Response · Authors · 2022-08-02
> **Largely improved writing and presentation; clarified some parts; explained how and why to choose the two planes to compute the rotation angle; added limitations and potential negative societal impact (2nd part)**
>
> > How and why to select the two planes to compute a rotation angle
>
> Introducing rotation angles is one of the main contributions of our method. It is important for conformer identification and helps to achieve global completeness as introduced in Sec 2.2 and Sec 2.3. Here we explain **how and why we select the two planes to compute rotation angle of an edge**.
> - In the Algorithm 1 in Appendix A.1, we show how to select the reference node $f_{i\backslash j}$ and $f_{j\backslash i}$ for $i$ and $j$. Specifically, $f_{i\backslash j}$ is $i$'s nearest neighboring node except $j$, and $f_{j\backslash i}$ is $j$'s nearest neighboring node except $i$. Then the rotation angle is computed based on the four nodes $f_{i\backslash j}, i, j, f_{j\backslash i}$.
> - In terms of the **selection strategy** (why we choose these two reference nodes, not the other two in Figure 1(b)), we think a selection strategy is valid as long as it is applied to every edge consistently. In this work, for an edge $e_{ij}$, we choose $f_{i\backslash j}$ and $f_{j\backslash i}$ as the reference nodes for $i$ and $j$, respectively. We apply this selection strategy to all edges in a 3D graph for forming two planes then computing rotation angles. **With such selection strategy, we can prove that our method is complete as shown in Sec 3.1. Besides, it is easier to prove completeness using this strategy as shown in Appendix A.3.** `We added some statements in Sec 2.2 to make it clear for readers.`
>
>
> **In addition, we explain why we only select a pair of planes to compute the rotation angle (not consider all pairs of planes).**
> - Firstly, let's extend the example in Figure 1(b) to a general case that $i$ has $p$ neighbors besides $j$, called $i_1, i_2, ..., i_p$,  and $j$ has $q$ neighbors besides $i$, called $j_1, j_2, ..., j_q$. Then for any neighbor node $i_n \in \{i_1, i_2, ..., i_p\}$ and $j_n \in \{j_1, j_2, ..., j_q\}$, we can compute a rotation angle based on the four nodes $i_n, i, j, j_n$. Thus we can compute $p\times q$ rotation angles for edge $ij$. **Most existing methods compute all of these rotation angles between any two planes like the references [1][18][30] in our paper.**
> - In our method, we select one neighbor node for $i$ and one neighbor node for $j$ to compute only one rotation angle for edge $ij$. Let's call these two selected planes as $plane_1$ and $plane_2$. **We show that with this rotation angle between $plane_1$ and $plane_2$, we can easily know all of the other rotation angles.** For example, the rotation angle between the other two planes $plane_3$ and $plane_4$ can be computed based on
> $angle(plane_3,plane_4)=angle(plane_1, plane_2) - angle(lpane_1, plane_3) + angle(plane_2, plane_4).$
> Here, $angle(plane_1, plane_2)$ is our computed rotation angle, $angle(lpane_1, plane_3)$ and $angle(plane_2, plane_4)$ can be easily obtained via our local completeness. **Therefore, computing one rotation angle is enough and complete without any information loss. We do not need to compute all rotation angles between any two planes.** Overall, our method is effective (complete) and much more efficient.
>
>
> > Limitations and potential negative societal impact
>
> Thanks for your reminder. **We added discussions of limitations and potential negative societal impact in Sec 7.**
>
> - One main **limitation**, not only for ComENet but also for existing 3D GNNs, is that existing 3D GNN models are designed **based on the fact that 3D information is given in data**. However **acquiring 3D information itself is difficult and expensive in practice**. Current methods rely on experiments or DFT-based computing, which is extremely time-consuming. Generative models and self-supervised methods are possible solutions to solve this problem.
> - ComENet can facilitate many important real-world applications, such as drug discovery and material discovery. It can be used in several research domains including quantum chemistry and physics, material sciences, molecular dynamics simulations, etc. **Any negative societal impact associated with those applications and domains can be applied to our method.**
>
> We sincerely thank you for your time! We look forward to your reply and further discussions, thanks!

---

> > ### Author Response · Authors · 2022-08-07
> > **Authors' follow-up**
> >
> > Dear Reviewer eaXa,
> >
> > Thank you again for your constructive comments and suggestions. And thanks for recognizing that our `soundness` and `contribution` are `good`. We have replied to your comments and revised our paper heavily to improve the `presentation` and make it clearer to readers. We hope that you could reply to our rebuttal and consider updating your score if we do have addressed your concerns. Also, please let us know if there are any additional concerns or feedback. Thank you!
> >
> > Sincerely,
> > Authors

---

> > ### Comment · Reviewer_eaXa · 2022-08-08
> > **Response**
> >
> > Thank you for the substantial explanation. Most of my concerns have been addressed and therefore I will increase my score.

---

> > > ### Author Response · Authors · 2022-08-08
> > > **Thank you**
> > >
> > > Thank you very much for your reply! And thanks again for your valuable comments and suggestions, which help improve our paper a lot!
> > >
> > > Sincerely,
> > > Authors

---

> ### Author Response · Authors · 2022-08-02
> **Largely improved writing and presentation; clarified some parts; explained how and why to choose the two planes to compute the rotation angle; added limitations and potential negative societal impact (1st part)**
>
> Many thanks for your valuable comments. **We have revised the manuscript heavily to improve the writing and presentation**, and we also provide responses here.
>
>
> > Clarify some parts
> - **$n$ and $k$**: `Existing methods exhibit excessive time complexity of
> $O(nk^2)$ or even $O(nk^3)$. But the complexity of our method is $O(nk)$.` Here $n$ and $k$ denote the number of nodes and the average degree in a 3D graph. **As you suggested, we revised the paper and reorganized several statements to define them before being used**. In the following part of our paper, we use the same definitions.
>
> - **$m$ and $d$**: `A geometric transformation maps a 3D graph to a geometric representation with size $m \times d$.` We updated $d$ to $h$ to avoid confusion with the notation for distance. Here $m$ is the number of transformed geometric features and $h$ is the feature size. $m$ and $h$ can be different for different methods as shown in this table. **We added a statement in the paragraph before Definition 1 to make it clear to readers.**
>
>     |Method|$m$|$h$|Why|
>     |----|----|----|----|
>     |SchNet|number of edges|1|SchNet computes distance $d_{ij}$ for each edge.|
>     |our ComENet|number of edges|4|ComENet computes distance and angles $d_{ij}, \theta_{ij}, \phi_{ij}, \tau_{ij}$ for each edge.|
>
> - **SE(3)** is Special Euclidean group in 3 dimensions. It includes arbitrary combinations of rotations and translations in 3D. Thus $R \in SE(3)$ is a transformation that combines rotation and translation. A rotation transformation can be represented with a $3 \times 3$ rotation matrix, and a translation transformation can be represented by a $3 \times 1$ vector. **SE(3)* is important in molecular learning**, and the reason why we introduce it here is straightforward: **When we rotate a 3D molecule, the coordinates of all nodes in the molecule would change, BUT it’s still the same molecule!** Translation is the same. We need to introduce **SE(3)** to try to avoid violating this nature. **We revised the paper and added several statements after Definition 1 to make it clear to readers.**
>
> - Yes, you are correct. **Strongly connected** is an important definition in graph theory. A graph is said to be strongly connected if every node is reachable from every other node. This means there is a path between any two arbitrary nodes. We are working on molecule representations, and all molecules in nature can be constructed as strongly connected graphs. **We revised the paper and added a statement after Proposition 1 to make it clear to readers.**
>
> - Thanks for the suggestion. **In Sec 4, we updated node feature vector** to **node feature vector $\textbf{v}$** in the text to make it clear and consistent with Figure 3.
>
> - **The definition of $\mathcal{L}_i$ and $\mathcal{L}_{ij}$**: The notations are consistent and used to explain the example in Figure 1(a). As shown in Figure 1(a), we define $\mathcal{L}_i$ as the 1-hop local structure centered in $i$, and $\mathcal{L}_{ij}$ as the 1-hop local structure centered in $j$. Here $j\in \mathcal{N}_i$ is any 1-hop neighbor of node $i$.
> Actually, in the original version of the paper, we have a statement `local structures centered in $i$ and $\mathcal{N}_i$, defined as $\mathcal{L}_{i}$ and $\mathcal{L}_{ij, j\in \mathcal{N}_i}$, respectively.` $j\in \mathcal{N}_i$ is just any 1-hop neighbor of node $i$. **To make it clearer, we added a statement in Sec 2.2 to specify an example for any $j$.**
>
> We continue at the 2nd part as below:

---

### Official Review · Reviewer_tzSo · 2022-07-11

**Rating:** 6
**Confidence:** 4
**Soundness:** 4 excellent
**Presentation:** 3 good
**Contribution:** 3 good

**Summary:**

This paper introduces a message-passing paradigm on 3D molecular graphs with better completeness (bijectivity) and additionally other advantages including operating within a 1-hop neighborhood, lower computational complexity, and faster training and inference. Analysis and experiments show that the proposed method can perform comparably well on multiple datasets with obviously increased efficiency.

**Questions:**

Questions:
- As "unless otherwise specified, the values for baseline methods are taken from the referred papers" (line 320), can I also assume that the (macro) network architectures for different baselines are not necessarily the same? I'd be interested in a comparison of the message-passing paradigm itself to other message-passing paradigms (e.g. the one in SphereNet[34]) under the same (macro) network architectures. If such a comparison is hard to conduct, some other details such as the network size of the baselines and ComENet may also help make the statements more convincing.

Suggestions:
- I may suggest first introducing the message passing scheme in Section 2.5 (e.g. Equation 1) and then showing its completeness (Section 2.2-2.4) -- reading the deductions with an overview of the final formulation and a summary of notations may help with faster comprehension. But just my two cents.

**Limitations:**

I probably missed it but I haven't found discussions of limitations and potential negative societal impact in the paper.

**Strengths And Weaknesses:**

Strengths:
- The paper brings up the problem of local completeness != global completeness which is kinda overlooked in prior works.
- The paper proposes a useful message-passing paradigm on molecular graphs with better completeness and efficiency.
- Systematical analysis and extensive experiments on multiple tasks under different evaluation metrics are provided.
- Good experimental results and analysis support the effectiveness of the proposed method.

Weaknesses:
- To my understanding, the main arguments of the paper are "completeness" and "efficiency" of the proposed method, and my feeling is that the latter is better addressed than the former in the experiments. As stated in Section 2.3, completeness is most important in identifying structures like conformers. More specific experiments on this (in addition to the general tasks) or some relevant details (e.g. how many conformers exist in the dataset) for the datasets could make the argument stronger, as a higher number on a generic task can be affected by many factors.

---

> ### Author Response · Authors · 2022-08-02
> **Added supporting experiments and analyses for completeness; compared different message-passing paradigms under the same network architecture; explained writing about Sec 2; added limitations and potential negative societal impact (3rd part)**
>
> > Writing suggestion
>
> Thanks for your suggestion! We totally agree that it is a good way to **firstly introduce an overview of the final formulation (Eq (1)) and a summary of notations and then showing completeness**. And our main delivery in Sec 2 is exactly the **message passing scheme defined in Eq (1) in Sec 2.5**.
>
> In our paper, we organize Sec 2 from another point of view. Specifically,
>   - **Sec 2.1** gives the definition of completeness.
>   - **Sec 2.2 and Sec 2.3** indicate $\tau$ is important for achieving global completeness.
>   - **Sec 2.4** tells why $d$, $\theta$, and $\phi$ are important for achieving local completeness.
>   - **Sec 2.5** gives our final message passing scheme considering all 3D geometric features.
>
> Overall, **we derive Eq (1) based on previous sections.** Therefore, we firstly introduce Sec 2.2, 2.3, and 2.4, then derive Sec 2.5 in current version. However, we do believe your suggestion is a good and clear alternative.
>
> > Limitations and potential negative societal impact
>
> Thanks for your reminder. We added discussions of limitations and potential negative societal impact in Sec 7.
>
> - One main **limitation**, not only for ComENet but also for existing 3D GNNs, is that existing 3D GNN models are designed **based on the fact that 3D information is given in data**. However **acquiring 3D information itself is difficult and expensive in practice**. Current methods rely on experiments or DFT-based computing, which is extremely time-consuming. Generative models and self-supervised methods are possible solutions to solve this problem.
> - ComENet can facilitate many important real-world applications, such as drug discovery and material discovery. It can be used in several research domains including quantum chemistry and physics, material sciences, molecular dynamics simulations, etc. **Any negative societal impact associated with those applications and domains can be applied to our method.**
>
> We sincerely thank you for your time! We look forward to your reply and further discussions, thanks!
>
> > Reference
>
> [1] Schütt, Kristof, Oliver Unke, and Michael Gastegger. "Equivariant message passing for the prediction of tensorial properties and molecular spectra." ICML 2021.
> [2] Klicpera, Johannes, Janek Groß, and Stephan Günnemann. "Directional message passing for molecular graphs." ICLR 2020.
> [3] Unke, Oliver T., and Markus Meuwly. "PhysNet: A neural network for predicting energies, forces, dipole moments, and partial charges." JCTC 2019.
> [4] Schütt, Kristof, et al. "Schnet: A continuous-filter convolutional neural network for modeling quantum interactions." NeurIPS 2017.
> [5] Gasteiger, Johannes, Florian Becker, and Stephan Günnemann. "Gemnet: Universal directional graph neural networks for molecules." NeurIPS 2021.

---

> > ### Comment · Reviewer_tzSo · 2022-08-09
> > **Reply**
> >
> > Dear authors,
> >
> > Thanks very much for your reply! I think my points are well-addressed. I really appreciate all the revisions and additional experiments that improve the paper. However, to my understanding, a score of 7 or above to some extent implies an inclination toward spotlight/oral acceptance, which I am usually very careful about.
> >
> > If I can see more strong arguments about the *novelty* of this paper, I may consider further raising my score to 7 -- but I would tend to be conservative on this.
> >
> > I'm also very sorry for my late reply.

---

> > > ### Author Response · Authors · 2022-08-09
> > > **Author’s follow-up for Reviewer tzSo**
> > >
> > > Dear Reviewer tzSo,
> > >
> > > Thank you very much for your reply! We are happy to know that `your points are well-addressed` in the revision and responses. We further summarize the novelty and impact of our paper here.
> > >
> > > > **Novelty**
> > >
> > > The main novelty of this paper is the design of a **complete and efficient** message passing for 3D molecular learning (can also be applied to other 3D structures in nature).
> > >
> > > - We propose a rigorous and provably complete method for 3D structures.
> > >   - It is guaranteed to incorporate 3D information **completely without information loss** (significant and accurate).
> > >   - It can **distinguish all molecular structures in nature**, and can identify conformers (powerful).
> > > - We provide novel strategies to achieve complete representations by considering both **local and global completeness**.
> > >   - We achieve global completeness with rotation angles. It can help identify conformers, which are finest molecular structures in nature. (as you suggested, we conducted substantial experiments to support it).
> > >   - We design to achieve local completeness with improved efficiency.
> > > - Our method is **much more efficient** than most recent 3D GNNs (like DimeNet, SphereNet) and is even at similar scale as 2D GNNs (don't consider 3D information).
> > >   - It can reduce complexity from $O(nk^2)$ to $O(nk)$ compared with previous methods like DimeNet and SphereNet. Here $n$ and $k$ denote the number of nodes and the average degree in a 3D graph. Note that the complexity for 2D GNNs is also $O(nk)$.
> > >   - Our experiments show that our ComENet can accelerate the training and inference by **6-10 times**.
> > >
> > > > **Impacts**
> > >
> > > - it is guaranteed to incorporate 3D information **completely without information loss**. It can **distinguish all molecular structures in nature**, and can identify conformers.
> > > - it is **efficient** and can be applied to large molecules and datasets. Results show that our ComENet can accelerate the training and inference by **6-10 times**.
> > >   - This is important because in modern **ML for science & engineering**, the scale of real-world data is becoming larger and larger. **Our ComENet can generate significant impacts because it makes the training of large-scale data feasible.**
> > >   - For example, considering the OC20 dataset by CMU and Fackbook AI, there are **134M** training samples. Our ComENet can reduce the training of S2EF from GPU years to several GPU weeks.
> > >   - In addition, as in [our discussion](https://openreview.net/forum?id=mCzMqeWSFJ&noteId=y7SQ_tKezy) with `Reviewer SSbD`, we can **directly apply our method to a protein dataset** with an average of 300 nodes in each graph (the largest number is 3615). Our result is very good compared to some baseline methods specifically designed for proteins. In addition, our method has only a small number of parameters and can be run using only one GPU. **This demonstrates the capability of our model on much larger macromolecules.**
> > >
> > > Given the above evidences (`complete representation, as efficient as 2D GNNs, great generalization to much larger macromolecules such as proteins, great potential in AI for science & engineering`), we feel strongly that our work is novel and deserves a score higher than 6. Meanwhile, we respect the reviewer's evaluation and judgement, as all scientific work is subject to peer and eventually community evaluations. Thanks!
> > >
> > > Sincerely,
> > > Authors

---

> ### Author Response · Authors · 2022-08-02
> **Added supporting experiments and analyses for completeness in identifying conformers; compared different message-passing paradigms under the same network architecture; explained writing about Sec 2; added limitations and potential negative societal impact (2nd part)**
>
> > **Network architectures** for different baselines are not necessarily the same, compare different message-passing paradigms under the same network architecture, show the number of parameters
>
> Thanks for your comments. Indeed, **existing baselines share the very similar macro network architecture**. For example, DimeNet [2] (Fig. 4) follows PhysNet [3] (Fig. 1) and they use very similar architecture, which consists of embedding, interaction, residual, and output blocks. Other studies like SchNet [4], GemNet [5], and PaiNN [1] also use similar architectures. **The different parts actually lie in their specific designs to adapt to their distinguishing message passing schemes.** Following the fashion in this research line, we also designed our network architecture based on previous studies like SphereNet. The interaction block aims to integrate the geometry information derived in the message passing, thus is the most important one in the whole architecture. We  used similar interaction block as SphereNet and replaced the $\Phi(d,\theta),\Phi(d,\theta, \phi)$ in SphereNet to our geometric features $TBF(d,\theta, \phi),SBF(d,\tau)$. To integrate our local and global completeness components, we then made some changes and designed two convolutions, namely, GlobalConv and LocalConv to incorporate the proposed geometric features. But since the message massing paradigm is different (previous studies operate within 2-hop but our message passing operates within 1-hop, which is the main reason why our method is much more efficient),  hence, input block and the output block were inevitably different between previous methods and ComENet.
>
> Overall, **due to the different message passing fashions, network architectures can not be exactly the same**. However, **we still think your concern is valid**. Hence, we followed your suggestions and implemented our general idea on the SphereNet backbone. Briefly, **we changed the SphereNet backbone a bit thus it can take our message passing and geometric features $TBF(d,\theta, \phi),SBF(d,\tau)$.** We denoted this model as `ComENet MP + SphereNet`. Results and the number of parameters on OC20 are listed in the following table. The results show that **when using the same model architecture (a little modification), our message passing paradigm still performs better than spherical message passing**. `We can add those results in the paper if you suggest so.`
>
> |Method|Num_para|MAE_1|MAE_2|MAE_3|MAE_4|MAE_ave|EwT_1|EwT_2|EwT_3|EwT_4|EwT_ave|
> |----|----|----|----|----|----|----|----|----|----|----|----|
> |`SphereNet`|2.8M|0.5632|0.6682|0.5590|0.6190|0.6023|4.56%|2.70%|4.59%|2.70%|3.64%|
> |`ComENet MP + SphereNet`|2.6M|0.5578|0.6854|0.5532|0.6012|0.5994|4.36%|2.70%|4.52%|2.65%|3.56%|
> |`ComENet`|4.2M|0.5558|0.6602|0.5491|0.5901|0.5888|4.17%|2.71%|4.53%|2.83%|3.56%|
>
>
> We continue at the 3rd part as below:

---

> ### Author Response · Authors · 2022-08-02
> **Added supporting experiments and analyses for completeness in identifying conformers; compared different message-passing paradigms under the same network architecture; explained writing about Sec 2; added limitations and potential negative societal impact (1st part)**
>
> Thanks a lot for your valuable comments! We have revised the manuscript accordingly and also provide responses here.
>
> > Supporting experiments and analyses for **completeness in identifying conformers**
>
> Thanks a lot for your suggestion. Firstly, we added more experiments and analyses on OC20 to show the effectiveness of completeness in identifying conformers. Then we conducted experiments on a new single molecular conformer-energy dataset to support our method. `We revised Sec 6.4 of the paper and added ablation study for identifying conformers.`
>
> - Analyses on OC20:
>
>     - OC20 dataset is a newly released dataset to model and discover catalysts. There are a total of 660,010 data samples for the IS2RE task we are working on. Each data sample is a combination of two parts, namely, adsorbate and catalyst. Among the 660,010 data samples, there are only 82 unique adsorbates and 11,500 unique catalysts. **Each adsorbate corresponds to different conformers in the dataset, and it is similar to the catalyst. Therefore, there exist different conformers of molecules in this dataset. We added Figure 5 to show the number of conformers for each adsorbate and catalyst in Sec 6.4.**
>     - We then did an additional experiment on OC20 to show the effectiveness of rotation angles $\tau$ since rotation angles are important in identifying structures like conformers. We remove the rotation angle $\tau$ from ComENet and denote it as `ComENet w/o $\tau$`. The results are listed in the following table (**also added in Sec 6.4 in the paper**). Here MAE_1 to MAE_4 and EwT_1 to EwT_4 are the energy MAE and EwT for four splits. Results show that **removing rotation angles $\tau$ can harm the performance of ComENet**. We believe such analysis results can well support the effectiveness of using $\tau$ to achieve global completeness thus identifying conformers in nature.
>
>         |Method|MAE_1|MAE_2|MAE_3|MAE_4|MAE_ave|EwT_1|EwT_2|EwT_3|EwT_4|EwT_ave|
>         |----|----|----|----|----|----|----|----|----|----|----|
>         |`ComENet`|0.5558|0.6602|0.5491|0.5901|0.5888|4.17%|2.71%|4.53%|2.83%|3.56%|
>         |`ComENet w/o $\tau$`|0.5585|0.6851|0.5574|0.6186|0.6049|4.13%|2.65%|4.13%|2.75%|3.42%|
>
> - Results on a molecular conformer-energy dataset:
>
>   Since there are two kinds of molecules in each data sample on OC20, we further conducted experiments on a single-molecule conformer-energy dataset.
>
>   **Following Sec 5.3.2 and Figure 4 of PaiNN [1]**, we did experiments of `ComENet` and `ComENet w/o $\tau$` with the same hyperparameter $r_{cut}=2.5\AA$ on **ferrocene dataset**. **The dataset is similar to the example in Figure 2 in our paper. The inputs are different conformers (with different rotation angles of Cp-Fe-Cp), and the outputs are corresponding rotational potential energies.** We compared our results with the curves in **Figure 4 of PaiNN**.
>     - For `ComENet`, our prediction can fit the target energy curve perfectly. The MAE on 1000 test data is 0.025. This means that our ComENet can distinguish different conformers and predict their corresponding energies.
>     - For `ComENet w/o $\tau$`, the result is bad and the predicted values are almost the same for different conformers. This means that the model cannot capture the difference between conformers.
>     - **These results show that our model can distinguish different conformers by considering rotation angles $\tau$, demonstrating the effectiveness of our complete representations.**
>
> We continue at the 2nd part as below:

---

> ### Author Response · Authors · 2022-08-08
> **Author’s follow-up to reviewer tzSo, one day before author-reviewer conversation ends**
>
> Dear Reviewer tzSo,
>
> Thanks again for your valuable comments and suggestions in your initial review, which helps improve our work a lot! Regarding your main concerns on **completeness in identifying conformers** and **comparison between message passing paradigms with the same model architecture**, we have conducted substantial experiments and also revised the paper heavily in our rebuttal on August 1st. Could you please check at your earliest convenience? Thanks!
>
> About supporting experiments and analyses for **completeness in identifying conformers**, we do agree this feedback is extremely important and critical in supporting our theory. Thanks again for your suggestion! As shown in our rebuttal and revised paper, we added both analyses on OC20 and an experiment on a molecular conformer-energy dataset to show ComENet's capability in identifying conformers. All of those can better support our theory that **global completeness helps identify conformers**.
>
> About **comparison between message passing paradigms with the same model architecture**, we conducted experiments on SphereNet architecture with some minor changes to fit our message passing. The results show that our message passing paradigm still performs better than spherical message passing on the same network architecture.
>
> We hope that you could reply to our rebuttal and consider raising your score if we do have addressed your concerns. Also, please let us know if there are any additional concerns or feedback. Thank you!
>
> Sincerely,
> Authors

---

### Official Review · Reviewer_h4eH · 2022-07-13

**Rating:** 5
**Confidence:** 3
**Soundness:** 2 fair
**Presentation:** 2 fair
**Contribution:** 2 fair

**Summary:**

The authors propose ComENet, a new network for molecular property prediction with 3D information. The authors define a new message passing way (see equation (1)) and a new network backbone Figure 3.

**Questions:**

Please refer to the ``Strengths And Weaknesses'' section.

**Limitations:**

Yes

**Strengths And Weaknesses:**

## Originality

1. The general framework is similar to PaiNN and DimeNet++, which is not novel to me.

2. In Eqn.(1), seems more features are used in mesage passing. The method is new but not surprising.


## Significance

1. The accuracy in Table 3 and 4 are not as good as baselines.

2. On OC20, the authors did not compare with the state-of-the-art algorithms.

3. If efficiency is the most important part of this paper, I am not sure whether we can directly apply Eqn(1) to existing backbones like DimeNet++?

## Clarify

1. Section 2: What are the key points to deliver? And how & why they help the models?

---

> ### Author Response · Authors · 2022-08-02
> **Novelty lies in our message passing; key points delivered in Sec 2 are complete and efficient message passing; baselines on OC20 & results on Molecule3D and QM9 analyzed (2nd part)**
>
> > What are the key points to deliver in Sec 2? And how & why do they help the models?
>
> In Sec 2, our main delivery is the **message passing scheme defined in Eq (1) in Sec 2.5**. The message passing in Eq (1) is fully complete and much more efficient than previous methods. Specifically, **Sec 2.2 and Sec 2.3 indicate why $\tau$ in Eq (1) is necessary. Sec 2.4 tells why $d$, $\theta$, and $\phi$ in Eq (1) are important.** We also show our message passing performs within 1-hop neighorhood in  Eq (1). **Overall, we derive Eq (1) directly based on previous sections.** It may look simple but significant because it’s complete (first work in the field) and efficient (as efficient as 2D GNNs). This is actually the main contribution of our paper. **All the used 3D features and message passing fashion in Eq (1) determine the model design in Sec 4.** For example, we should use all four features in Fig 3; we use GlobalConv to handle $\tau$ and LocalConv to deal with $(d, \theta, \phi)$, corresponding to Sec 2.2 and Sec 2.4, respectively; our graph Conv operations in Fig 3 operate within 1-hop neighborhood, etc.
>
> > Baseline methods on OC20, results on Molecule3D and QM9
>
> - **Baseline methods on OC20**
>   - Firstly, as shown in Sec 5 that is related work, we categorize methods for 3D graphs into three directions, namely, equivariant 3D GNNs, invariant 3D GNNs, and methods with both absolute (equivariant) and relative (invariant) 3D information. **Our method belongs to invariant 3D GNNs** by only using relative information as input to avoid the limitations of equivariant 3D GNNs. Therefore, **we mainly compare our method with invariant 3D GNNs** like SchNet, DimeNet++, SphereNet, and GemNet. The performace of equivariant 3D GNNs is usually worse.
>   - Secondly, as shown in the first paragraph of Sec 6.1, we focuse on the IS2RE task, aiming to predict relaxed energy **directly from the initial structure**. Therefore, we only use the initial structure information and **do not compare with methods using extra information for fair comparison**. Specifically, **we do not compare with methods using relaxation, trajectory information, or relaxed structures, because these methods essentially use much more extra data** Using relaxation is computationally expensive during prediction while the relaxation trajectory and relaxed structures are hard to obtain in practice. **Methods like SCN, GemNet, 3D-Graphormer, and GNS + Noisy Nodes in the [Leaderboard](https://opencatalystproject.org/leaderboard.html)** either use the relaxation method or used relaxed structures. Therefore, we do not compare with them.
>
> - **Results on Molecule3D and QM9**
> In short, our ComENet either performs much better or converges much faster compared with baseline methods on Molecule3D and QM9. Besides, our results are comparable with the SOTA methods.
>   -  **Performance.** On both Molecule3D and QM9, our results are comparable with the SOTA methods. Specifically, on Molecule3D, our ComENet performs much better than GIN_virtual and SchNet, but a little worse than DimeNet++ and SphereNet. On QM9, our ComENet can outperform all baseline methods except SphereNet. **The little worse performance may be due to the relatively small molecules in Molecule3D (the average number of nodes per graph is 29) and QM9 (the average number of nodes per graph is 18). The structures that our complete strategy can distinguish may not exist in the dataset.** The statistics of the datasets are shown in Table 1. Our method shows superior power on large molecules like in OC20 data.
>   -  **Efficiency.** In addition, our method is much more efficient than the SOTA methods like DimeNet++ and SphereNet. As shown in Figure 4, ComENet dramatically reduces training time by 6-9 times compared with DimeNet++ and SphereNet, and costs similar time as GIN-virtual and SchNet.
>
> We sincerely thank you for your time! We look forward to your reply and further discussions, thanks!
>
> [1] Schütt, Kristof, et al. "Schnet: A continuous-filter convolutional neural network for modeling quantum interactions." NeurIPS 2017.
> [2] Klicpera, Johannes, Janek Groß, and Stephan Günnemann. "Directional message passing for molecular graphs." ICLR 2020.
> [3] Liu, Yi, et al. "Spherical message passing for 3d molecular graphs." ICLR 2022.
> [4] Gasteiger, Johannes, Florian Becker, and Stephan Günnemann. "Gemnet: Universal directional graph neural networks for molecules." NeurIPS 2021.
> [5] Schütt, Kristof, Oliver Unke, and Michael Gastegger. "Equivariant message passing for the prediction of tensorial properties and molecular spectra." ICML 2021.
> [6] Unke, Oliver T., and Markus Meuwly. "PhysNet: A neural network for predicting energies, forces, dipole moments, and partial charges." JCTC 2019.

---

> > ### Comment · Reviewer_h4eH · 2022-08-08
> > **Reply**
> >
> > Thanks for the response.
> >
> > 1. I did not hand on QM9 but I think the data split (how to obtain train/valid/test) sets are super important. Please ensure you use the same split as previous work.
> >
> > 2. "Novelty lies in our message passing" => Thanks for the reply. I am not sure whether the novelty is enough for NeurIPS. I hope AC and other reviewers can double check the comments.
> >
> > 3. Results on OC20: not convincing to me. Considering that "Novelty lies in message passing", it should be generally good.

---

> > > ### Author Response · Authors · 2022-08-08
> > > **Author’s follow-up for Reviewer h4eH**
> > >
> > > Dear Reviewer h4eH,
> > >
> > > Thank you very much for your reply. Since we have addressed most of the concerns about `originality, significance, and clarify` in your original comments and you also think `it should be generally good`, could you consider updating your score? Thank you very much.
> > >
> > > Here are our responses to your following comments:
> > >
> > > > **Novelty lies in our message passing**:
> > >
> > > The main novelty of this paper is the **design of a provable complete and efficient message passing in Sec 2&3**, and this is recognized by the other three reviewers. All other three reviewers think our `contribution` is `good` with a rating of `6`. For example, Reviewer `tzSo` says
> > > `the paper brings up the problem of local completeness != global completeness, proposes a useful message-passing paradigm on molecular graphs with better completeness and efficiency`, Reviewer `eaXa` says `instead of the intuition, the paper defines the global completeness and provides theoretical ground for the model to capture the global completeness`.
> > >
> > > > QM9, the same split
> > >
> > > Thanks for your reminder. For **QM9**, we compare with methods including SchNet, PhysNet, MGCN, PaiNN, DimeNet, DimeNet++, and SphereNet. **All of these baseline methods use 110000/10000/10831 for train/valid/test splits.** Specifically,
> > > - we use exactly the same data splits as DimeNet, DimeNet++, and SphereNet with the random split seed of 42.
> > > - the results of old baselines PyhsNet and MGCN are taken from recent work DimeNet [1] and DimeNet++ [2]. The results of SchNet are taken from the original paper [3] and are consistent in [1] and [2]. These methods use different random split seeds. Given 1). the data distribution is uniform thus the difference is minimal and  2). **their results are MUCH worse than DimeNet, DimeNet++, SphereNet, and our ComENet**, we don't reimplement these methods. We revised Sec 6.3 of our paper to clarify this.
> > > - PaiNN [4] is a recent work published in ICML 2021, and also achieves good performance. It also uses 110000/10000/10831 for train/valid/test splits, but with 3 random splits. Mean and Std are reported. Therefore, we re-ran our ComENet over 3 random splits on two properties, including $\alpha$ and $\epsilon_\text{HOMO}$. ComENet is similar with PaiNN on $\alpha$, and is better on $\epsilon_\text{HOMO}$. We revised our paper and added these results in Sec 6.3 for clarification.
> > >
> > >   |Property|PaiNN|ComENet|ComENet (3 runs)|
> > >   |---|---|---|---|
> > >   |$\alpha$|0.045|0.0452|$0.045\pm0.000$|
> > >   |$\epsilon_\text{HOMO}$|27.6|23.1|$23.2\pm0.2$|
> > >
> > > The results over 3 runs are similar to our original paper. This shows that the effects of different random seeds are minimal, which may be due to the fact that the QM9 dataset is stable in distribution.
> > >
> > >
> > > > Baseline methods for **OC20**
> > >
> > > Thanks for your question.
> > > - All of the methods in Table 2 use the same dataset with **460,328** training samples. We consider these methods as baselines for fair comparisons. ComENet is better than all baselines on energy MAE and is slightly worse than SphereNet on EwT (for sure, ComENet is much more efficient, and it can accelerate training&inference by 6-10 times.)
> > > - Some methods in the leaderboard [5] can achieve better results, but they **use additional information**. For example, `GemNet-T EFwT-Relaxation-All` can achieve the average energy MAE of 0.4, but use the large S2EF data with **134M** samples, which is about **300 times larger than our used training data**. **The inference is about 200 times more expensive** [6]. Therefore, we don't compare these methods. In addition, the task (and setting) we are working on `is the most common task in catalysis` [6] which is a more useful task in practice for catalyst discovery.
> > > - Besides, even for the large S2EF data, our method is much more efficient than existing methods like SphereNet and DimeNet++. It can reduce the training of S2EF from GPU years [7] to several GPU weeks. We do believe this is important because in modern ML for science, the scale of real-world data is becoming larger and larger. Our ComENet can generate significant impacts because it is complete (accurate) and it also makes the training of large-scale data feasible.
> > >
> > > Thank you again for your constructive comments and suggestions. We hope that you could consider updating your score if we do have addressed your concerns. Also, please let us know if there are any additional concerns or feedback. Thank you!
> > >
> > > Sincerely,
> > > Authors
> > >
> > > [1] Directional message passing for molecular graphs
> > >
> > > [2] Fast and Uncertainty-Aware Directional Message Passing for Non-Equilibrium Molecules
> > >
> > > [3] SchNet: A continuous-filter convolutional neural network for modeling quantum interactions
> > >
> > > [4] Equivariant message passing for the prediction of tensorial properties and molecular spectra
> > >
> > > [5] https://opencatalystproject.org/leaderboard.html
> > >
> > > [6] The Open Catalyst 2020 (OC20) Dataset and Community Challenges
> > >
> > > [7] ForceNet: A Graph Neural Network for Large-Scale Quantum Calculations

---

> > > > ### Comment · Reviewer_h4eH · 2022-08-09
> > > > **Reply**
> > > >
> > > > Thanks for your reply.
> > > >
> > > > Thanks for the clarification. Please release the code for reproducibility in the future.
> > > >
> > > > I'd like to increase my score to five.

---

> > > > > ### Author Response · Authors · 2022-08-09
> > > > > **Thank you**
> > > > >
> > > > > Thank you very much for your reply! We are happy to know that your concerns were addressed in the revision and responses. In addition, we will release the code for reproducibility after the anonymous review period. Thanks again for your valuable comments and suggestions!
> > > > >
> > > > > Sincerely,
> > > > > Authors

---

> ### Author Response · Authors · 2022-08-02
> **Novelty lies in our message passing; key points delivered in Sec 2 are complete and efficient message passing; baselines on OC20 & results on Molecule3D and QM9 analyzed(1st part)**
>
> Thanks a lot for your constructive comments! We have revised the manuscript accordingly and also provide responses here.
>
> > The Originality that the general framework is similar to PiaNN and DimeNet++
>
> - Firstly, we argue that **the main novelty of this paper is the design of a complete and efficient message passing in Sec 2&3**. **This determines what information we should use and how to use it in the framework part in Sec 4.** Regarding content and space, the message passing part (design, proofs) occupies pages 2-5. The network part only occupies page 6 to state how to integrate the message passing in the network using standard components like graph convolution, MLP, nonlinearities, etc. Our main novelty is also recognized by the other three reviewers.
> - Secondly, for all **existing 3D GNNs** (SchNet [1], DimeNet [2], SphereNet [3], GemNet [4], PaiNN [5], etc), the major different parts lie in different message passing methods. They just **share a very similar framework**. Starting from SchNet, important papers in this research line always use a similar pipeline, which roughly consists of input blocks, interaction blocks, and output blocks. For example, DimeNet (Fig. 4) follows PhysNet [6] (Fig. 1) and they use very similar architecture, which is consist of embedding, interaction, residual, and output blocks. Other studies like SchNet, GemNet, and PaiNN also use similar architectures. **Each of the methods makes a distinguishing contribution to the community own to its novel message passing method for incorporating 3D information.**
> - Thirdly, even for framework, our **ComENet designs a most different one** compared with existing methods, because we use the Self-Atom layer and propose two specifically designed graph conv techniques including **LocalConv and GlobalConv**, to better implement our message passing derived in Sec 2 and proved in Sec 3.
> - **We revised Sec 4 of the paper to make this clearer to readers**.
>
> > Key equation (Eq 1) uses more features in message passing
>
> **In short, Eq (1) indicates both completeness and effciency, which are main contributions of our work. Eq (1) contains all 3D features that are necessary to achieve full completeness, and Eq (1) also indicates these features are computed within 1-hop neighborhood, leading to high efficiency.**
> - Firstly, the motivation of this work is **NOT simply using more 3D features**, but to conduct rigorous analysis to see what 3D features would we use to achieve **completeness**, and how should we use them **efficiently**. Existing methods use different types of features, like SchNet uses distances, DimeNet further uses angles, and SphereNet further uses torsions. **Adding more features leads to better performance, which is not surprising.** Then a fundamental research problem in the field could be: **what 3D features should we use so that the 3D information is fully harnessed, then any 3D structure in nature can be determined and distinguished.** `This essentially motivates our work.` Then in our work, we conduct rigorous and provable analysis to show that **by using our derived features in Eq (1), we can incorporate 3D information completely**. Thus our methods can distinguish any different structures in nature, even the finer structures like conformers for the same molecule.
> - In addition, we wisely design the message passing strategy in Eq (1) (**message passing operates within 1-hop neighborhood as in Eq (1)**), leading the complexity of **O(nk)**. However, existing methods like DimeNet++ and SphereNet have the complexity of **O(nk^2)**. Obviously, our method is **more scalable in large-scale or industry-scale applications**. We believe the texts before and after Eq (1) can also help elaborate on the significance of Eq (1).
>
> > Apply Eq (1) to existing backbones like DimeNet++ to achieve efficiency.
>
> There’s some misunderstanding here, because **Eq (1) can not be applied to other backbones like DimeNet++**. Eq (1) is the main finding in Sec 2, it then determines how we design the model architecture in Sec 4. Apart from completeness, we show that **Eq (1) performs node-update message passing fashion within 1-hop neighborhood**, which is more efficient. Accordingly, all the features in Eq (1) are also computed based on 1-hop neighborhood (also in Algorithm 1 in Appendix). **Existing studies including DimeNet++ and SphereNet use edge-update message passing performing within 2-hop neighborhood.** Hence, Eq (1) cannot be applied directly to them. Instead, **we specifically design the network in Sec 4 to better adapt to Eq (1)**.
>
> `We thank the reviewer for the question, and we revised the paper and added more details at the end of Sec 2.5 to clarify this.`
>
> We continue at the 2nd part as below:

---

### Author Response · Authors · 2022-08-06
**To all reviewers: Could you please reply to our rebuttal and consider updating your scores?**

Dear Reviewers,

Thank you for your constructive comments and suggestions. Our point-by-point responses can be found below. We also have revised our paper heavily based on your suggestions and comments. Since it is approaching the end of the Author-Reviewer Discussion, we hope that you could reply to our rebuttal and consider updating your scores if we do have addressed your concerns. Also, please let us know if there are any additional concerns or feedback. Thank you!

Sincerely,

Authors

---

### Meta-Review · Area_Chair_WkhH · 2022-08-25

**Recommendation:** Accept
**Confidence:** Certain

**Metareview:**

This paper presents useful MP for molecular graphs with better completeness properties.
I encourage the authors to consider addressing the reviewers' comments while preparing their revision.
In particular, discuss relation to previous work such as PaiNN, DimeNet++. Clarify splits. Consider adding "completeness". Address exposition issues (eaXa, SSbD).

**Award:**

No

---

### Decision · Program_Chairs · 2022-09-14

Accept